# A New HIV-1 $K_{28}E_{32}$-Reverse Transcriptase Variant Associated with the Rapid Expansion of CRF07_BC among Men Who Have Sex with Men

Jingwan Han,[a] Yan-Heng Zhou,[b,c,d] Yingying Ma,[b] Guoxin Zhu,[a] Dong Zhang,[a] Bo Zhu,[a] Tong Cheng,[e] Lanfeng Wang,[e] Jian-Hua Wang,[d] Lin Li,[a] Chiyu Zhang[b]

[a]Department of AIDS Research, State Key Laboratory of Pathogen and Biosecurity, Beijing Institute of Microbiology and Epidemiology, Beijing, China
[b]Shanghai Public Health Clinical Center, Fudan University, Shanghai, China
[c]Shaanxi Engineering and Technological Research Center for Conversation and Utilization of Regional Biological Resources, College of Life Sciences, Yan'an University, Yan'an, Shaanxi, China
[d]Guangzhou Institutes of Biomedicine and Health, Chinese Academy of Sciences, Guangzhou, China
[e]Institut Pasteur of Shanghai, Chinese Academy of Sciences, Shanghai, China

Jingwan Han and Yan-Heng Zhou contributed equally to this article. Author order is dependent on their contributions to experiments.

**ABSTRACT** HIV-1 CRF07_BC originated among injection drug users (IDUs) in China. After diffusing into men who have sex with men (MSM), CRF07_BC has shown a rapid expansion in this group; however, the mechanism remains unclear. Here, we identified a new $K_{28}E_{32}$ variant of CRF07_BC that was characterized by five specific mutations (E28K, K32E, E248V, K249Q, and T338S) in reverse transcriptase. This variant was mainly prevalent among MSM, and was overrepresented in transmission clusters, suggesting that it could have driven the rapid expansion of CRF07_BC in MSM, though founder effects cannot be ruled out. It was descended from an evolutionary intermediate accumulating four specific mutations and formed an independent phylogenetic node with an estimated origin time in 2003. The $K_{28}E_{32}$ variant was demonstrated to have significantly higher *in vitro* HIV-1 replication ability than the wild type. Mutations E28K and K32E play a critical role in the improvement of *in vitro* HIV-1 replication ability, reflected by improved reverse transcription activity. The results could allow public health officials to use this marker (especially E28K and K32E mutations in the reverse transcriptase (RT) coding region) to target prevention measures prioritizing MSM population and persons infected with this variant for test and treat initiatives.

**IMPORTANCE** HIV-1 has very high mutation rate that is correlated with the survival and adaption of the virus. The variants with higher transmissibility may be more selective advantage than the strains with higher virulence. Several HIV-1 variants were previously demonstrated to be correlated with higher viral load and lower CD4 T cell count. Here, we first identified a new variant (the $K_{28}E_{32}$ variant) of HIV-1 CRF07_BC, described its origin and evolutionary dynamics, and demonstrated its higher *in vitro* HIV-1 replication ability than the wild type. We demonstrated that five RT mutations (especially E28K and K32E) significantly improve *in vitro* HIV-1 replication ability. The appearance of the new $K_{28}E_{32}$ variant was associated with the rapidly increasing prevalence of CRF07_BC among MSM.

**KEYWORDS** HIV-1, CRF07_BC, variant, reverse transcriptase, men who have sex with men, replication ability, transmission cluster, human immunodeficiency virus

Address correspondence to Chiyu Zhang, chiyu_zhang1999@163.com, or Lin Li, dearwood@sina.com.

The authors declare no conflict of interest.

Since the first case was reported in 1985, HIV/AIDS has been a national problem in China, with 1,045,000 people living with HIV/AIDS by the end of 2020 (1). China has experienced several large changes in HIV-1 epidemic since 1985 (2). First, the major HIV transmission routes shifted from blood transmission via injection drug use (IDU)

and illegal blood donation to sexual transmission, especially homosexual transmission among men who have sex with men (MSM) (2, 3). Second, the genetic diversity of HIV-1 rapidly increased with on-going generation of new circulating recombinant forms (CRFs) and various unique recombinant forms (URFs) (4, 5). Third, the predominant HIV-1 subtypes have switched from B, C, and CRF01_AE in the 1990s to CRF01_AE, CRF07_BC, and B most recently (5–7). The rise of CRF07_BC has raised large concern.

CRF07_BC originated in early 1990s, and mainly circulated among injection drug users (IDUs) (8, 9). After diffusing into heterosexual and homosexual transmission networks, CRF07_BC rapidly increased in prevalence (5, 6, 10, 11). Currently, it accounts for 20.5% of all subtyped infections in China, and is now the second most predominant HIV-1 strain, following CRF01_AE (39.7%) (5). This growth coincided with an increase in HIV incidence among MSM (12). CRF07_BC has mostly replaced CRF01_AE as the most predominant HIV-1 strain among MSM since 2010 (11, 13). Why CRF07_BC is rapidly expanding among MSM remains unclear. Recently, CRF07_BC was demonstrated to have enhanced transmission capability over subtype B and CRF01_AE, which might be associated with a 7 amino acid deletion in the p6 region of the Gag protein (p6Δ7) (14). However, the p6Δ7 variant did not explain the rapidly growing prevalence of CRF07_BC among MSM since it originated among IDUs and was prevalent among both IDUs and MSM (15, 16).

HIV-1 is mostly spread along contact networks with sexual or blood exposure risks (17, 18). Network analysis provides a robust tool to understand HIV-1 transmission over space and time and allows characterization of sequence features associated with large transmission networks (19). Here, we identified a new variant of HIV-1 CRF07_BC using transmission network analysis, and reported its origin and evolutionary history. The new variant known as the $K_{28}E_{32}$ variant was characterized by 5 specific amino patterns (Lys [K], Glu [E], Val [V], Glu [Q] and Ser [S], respectively) at sites 28, 32, 248, 249, and 338 of reverse transcriptase (RT) coding region and was demonstrated to have higher *in vitro* HIV-1 replication activity than the wild type. Very high prevalence of the $K_{28}E_{32}$ variant among MSM and its overrepresentation in transmission clusters suggest that its appearance was associated with the rapid expansion of CRF07_BC among MSM.

## RESULTS

**Identification of the $K_{28}E_{32}$ variant of HIV-1 CRF07_BC.** Based on the phylogenetic analysis of CRF07_BC *RT* coding region sequences (2289-3187nt in HXB2) from 1997–2013, eight large evolutionary (or transmission) clusters (ECs or TCs) were identified, consisting of 510 sequences (42.7%) (Fig. S1). There were 350 (29.3%) non-cluster sequences, and the remaining 335 sequences formed small transmission clusters with <20 sequences. Of the 8 large ECs, 5 contained the sequences ($n = 383$) obtained from 2007 to 2013, and were named post-2007 clusters. The other 3 ECs included 127 sequences obtained during 1997 to 2012, and were named pre-2007 clusters. We then investigated whether there was a difference in signature residues between cluster and non-cluster sequences. We found 2 distinct amino acid sequence features that separated cluster and non-cluster sequences. The vast majority of cluster sequences had Lys (K) and Glu (E) residues at sites 28 and 32 of RT coding region (376/510: 73.7%) (https://www.hiv.lanl.gov/content/sequence/LOCATE/locate.html), respectively, and were defined as the $K_{28}E_{32}$ variant, while the vast majority of non-cluster sequences (269/350: 76.9%) had E and K residues at 28 and 32 sites, respectively, and were defined as the wild-type (WT) or the $E_{28}K_{32}$ strain (chi-square test, $P < 0.001$) (Fig. 1a). Interestingly, all 5 post-2007 ECs carried the $K_{28}E_{32}$ variant, and the pre-2007 clusters contain the wild-type strains (Fig. 1a).

Because of only 898 nt of RT coding region was included in above analysis, we further investigated whether the $K_{28}E_{32}$ variant carried other specific amino acids in RT using all available sequences of the entire CRF07_BC RT coding region. We found that the vast majority (92.4%) of the $K_{28}E_{32}$ variants carried 3 additional specific amino acids mutations E248V, K249Q, and T338S in the RT coding region

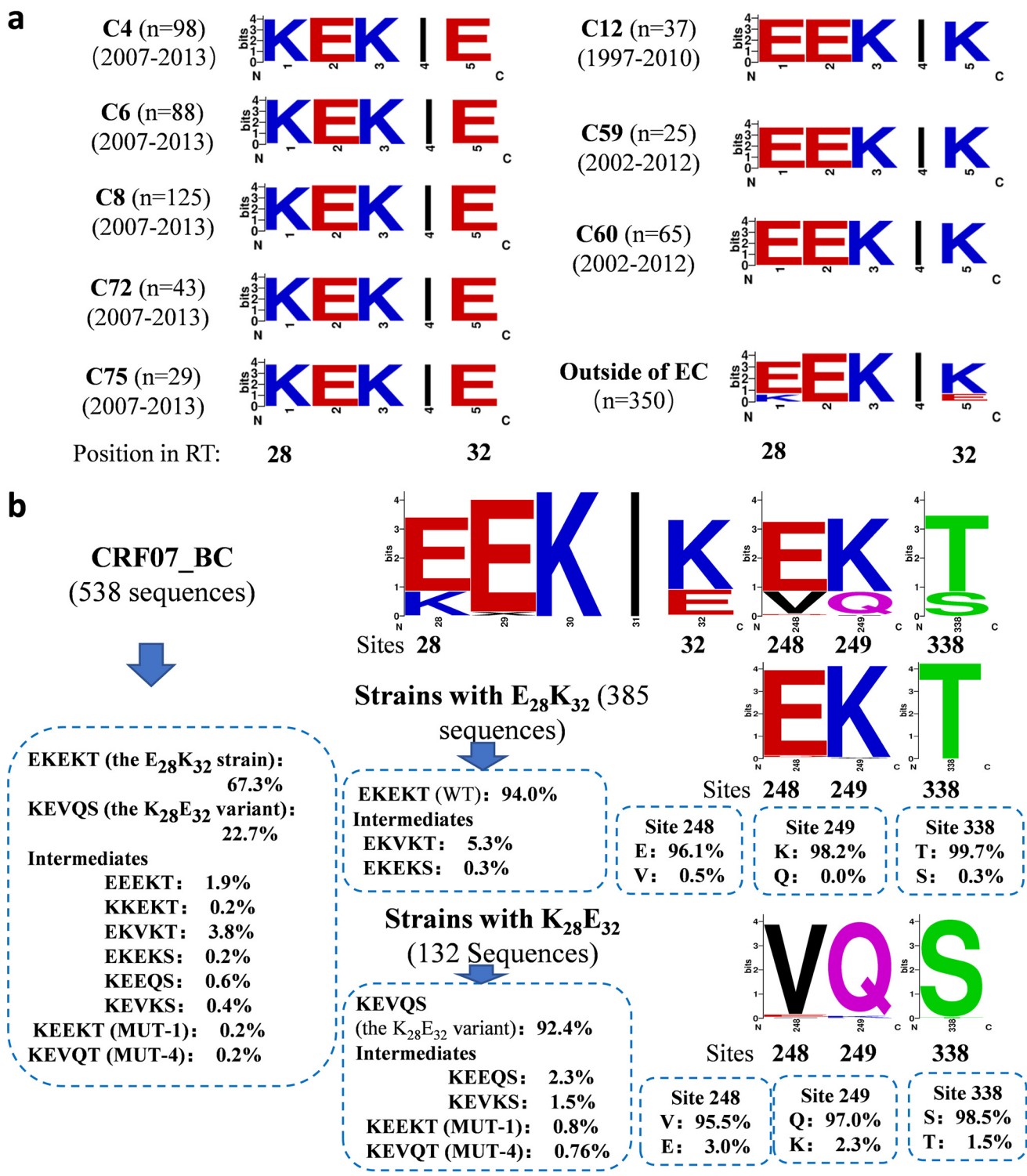

**FIG 1** RT coding sequence characteristics of the $K_{28}E_{32}$ variant and wild-type of HIV-1 CRF07_BC. (a) Sequence characteristics of CRF07_BC strains within and outside of transmission clusters in the preliminary ML analysis (shown in Fig. S1). The location of amino acids was based on the RT coding region of HXB2 strain. The numbers of sequences and their sampling years were shown in parentheses. C, cluster or evolutionary cluster (EC). (b) Amino acid characteristics of CRF07_BC at sites 28, 32, 248, 249 and 338 of RT. The percentages of the $K_{28}E_{32}$ variant, wild-type and the intermediates are shown.

(Fig. 1b). Therefore, the $K_{28}E_{32}$ variant was featured by K-E-VQ-S at 28, 32, 248/249 and 338 sites of RT coding region, respectively, while the wild-type strain by E-K-EK-T. The $K_{28}E_{32}$ variant accounted for about 22.7% of all analyzed sequences (Fig. 1b).

**FIG 2** Amino acid characteristics at the 5 special sites of RT coding region. Sites: 28, 32, 248/249, 338.

| Sequence | 28 32 (248/249) 338 |
|---|---|
| **Wild type** | |
| CRF07_BC. CN. 05. XJDC6431_2. EF368372 | EEKIK  EK  T |
| CRF07_BC. CN. 98. 98CN009. AF286230 | .....  ..  . |
| **K₂₈E₃₂ variant** | |
| CRF07BC_SZ_2013_LS6804_IDU | K...E  VQ  S |
| CRF07BC_CN_2007_GZ070087_KF250372_MSM | K...E  VQ  S |
| A1. RW. 92. 92RW008. AB253421 | .....  D.  . |
| A2. CM. 01. 01CM_1445MV. GU201516 | .....  ..  . |
| B. FR. 83. HXB2_LAI_IIIB_BRU. K03455 | .....  ..  . |
| B. US. 98. 1058_11. AY331295 | .....  ..  . |
| C. IN. 95. 95IN21068. AF067155 | .....  ..  . |
| C. ZA. 04. 04ZASK146. AY772699 | .....  ..  . |
| D. CD. 83. ELI. K03454 | .....  ..  . |
| D. TZ. 01. A280. AY253311 | .....  ..  . |
| F1. FR. 96. 96FR_MP411. AJ249238 | .....  ..  . |
| F2. CM. 97. CM53657. AF377956 | .....  D.  . |
| G. BE. 96. DRCBL. AF084936 | .....  N.  . |
| G. PT. x. PT2695. AY612637 | .....  D.  . |
| H. BE. 93. VI991. AF190127 | .....  ..  . |
| H. CF. 90. 056. AF005496 | .....  ..  . |
| J. SE. 93. SE9280_7887. AF082394 | .....  ..  . |
| K. CD. 97. 97ZR_EQTB11. AJ249235 | .....  D.  . |
| K. CM. 96. 96CM_MP535. AJ249239 | .....  D.  . |
| L. CD. 01. L_CG_0018a_01. MN271384 | .....  D.  S |
| L. CD. 90. 90CD121E12. AF457101 | .....  D.  S |
| CRF62_BC. CN. 10. YNFL13. KC870034 | .....  ..  . |
| CRF62_BC. CN. 10. YNFL18. KC870037 | .....  ..  . |
| CRF64_BC. CN. 09. 09YNLX219037sg. KC899009 | .....  ..  . |
| CRF64_BC. CN. 10. YNFL10_1. KC870032 | .....  ..  . |
| CRF65_cpx. CN. 10. YNFL01. KC870027 | .....  ..  . |
| CRF65_cpx. CN. 11. ANHUI_HF104. KC183778 | K...E  ..  . |
| CRF67_01B. CN. 11. ANHUI_HF115. KC183779 | .....  ..  . |
| CRF67_01B. CN. 11. ANHUI_MAS59. KC183780 | .....  ..  . |
| CRF68_01B. CN. 10. JS2010001. KF758551 | .....  ..  . |
| CRF68_01B. CN. 11. ANHUI_XC46. KC183783 | .....  ..  . |
| CRF78_cpx. CN. 13. YNTC19. KU161143 | .....  ..  . |
| CRF78_cpx. CN. 13. YNTC88. MN654145 | .....  ..  . |
| CRF79_0107. CN. 15. SX15DT013. KY216146 | .....  ..  . |
| CRF79_0107. CN. 15. SX15JC12. KY216148 | .....  ..  . |
| CRF80_0107. CN. 11. YA285. MH843712 | K...E  VQ  . |
| CRF80_0107. CN. 12. YA376. MH843713 | K...E  VQ  . |
| CRF85_BC. CN. 14. 14CN_SCYB12. KU992936 | .....  ..  . |
| CRF85_BC. CN. 14. 14CN_SCYB3. KU992931 | .....  ..  . |
| CRF86_BC. CN. 13. 15YNHS18. KX582249 | .....  ..  . |
| CRF86_BC. CN. 13. 15YNHS26. KX582251 | .....  ..  . |

| Sequence | 28 32 (248/249) 338 |
|---|---|
| CRF07_BC. CN. 05. XJDC6431_2. EF368372 | EEKIK  EK  T |
| CRF87_cpx. CN. 09. 09YNLC497sg. KC898992 | .....  ..  . |
| CRF87_cpx. CN. 12. DH32. KF250408 | .....  ..  . |
| CRF88_BC. CN. 05. 05YNRL07sg. KC898975 | .....  ..  . |
| CRF88_BC. CN. 09. DH19. KF250402 | .....  ..  . |
| CRF96_cpx. CN. 10. JL. RF01. KF850149 | .....  ..  . |
| CRF96_cpx. CN. 13. 13YNBS66IDU. MG518477 | .....  ..  . |
| CRF100_01C. CN. 13. YNLC27. MH909568 | .....  ..  . |
| CRF100_01C. CN. 13. YNLC30. MH909570 | .....  ..  . |
| CRF101_01B. CN. 07. 07CNYN370. KF835546 | .....  ..  . |
| CRF101_01B. CN. 13. YNZT036. MK158945 | .....  ..  . |
| CRF102_0107. CN. 17. FY058. MN178644 | K...E  VQ  S |
| CRF102_0107. CN. 18. FY336. MN178645 | K...E  VQ  S |
| CRF103_01B. CN. 18. HE18S0290. MN067222 | .....  ..  . |
| CRF103_01B. CN. 18. HE18S0322. MN067224 | .....  ..  . |
| CRF104_0107. CN. 15. M62. MH396608 | K...E  ..  . |
| CRF104_0107. CN. 18. MSM20183420. MK564326 | K...E  ..  . |
| CRF105_0108. CN. 14. XC2014EU01. KX353919 | .....  ..  . |
| CRF105_0108. CN. 18. LS18S0127. MN752128 | .....  ..  . |
| CRF106_cpx. CN. 15. YN15099. MT277001 | .....  .Q  . |
| CRF106_cpx. CN. 18. LC18S083. MT276999 | .....  .Q  . |
| CRF107_01B. CN. 18. HL18S17. MT712388 | .....  ..  . |
| CRF107_01B. CN. 18. HL18S244. MT712390 | .....  ..  . |
| CRF109_0107. CN. 14. LS11584. MT919517 | .....  VQ  . |
| CRF109_0107. CN. 14. LS14250. MT919518 | .....  VQ  . |
| CRF110_BC. CN. 07. 07CNYN338. KF835524 | .....  ..  . |
| CRF110_BC. CN. 12. YN10189F. MW419275 | .....  ..  . |
| CRF111_01C. CN. 16. 16YN29. MT624751 | .....  ..  . |
| CRF112_01B. CN. 18. 18110456. MW018130 | .....  ..  . |
| CRF112_01B. CN. 19. BL4450_00. MW018137 | .....  ..  . |
| CRF113_0107. CN. 17. BL3128_00. MW018132 | K...E  VQ  . |
| CRF113_0107. CN. 19. BL3958_00. MW018136 | K...E  VQ  . |
| CRF114_0155. CN. 18. A18003. MN654104 | ....I  ..  . |
| CRF114_0155. CN. 19. VCT19012. MN654109 | .....  ..  . |
| CRF115_01C. CN. 12. kang140_NFL. KJ778896 | K...E  ..  . |
| CRF115_01C. CN. 13. kang019a_NFL. KJ778895 | K...E  ..  . |
| CRF116_0108. CN. 14. 14YN263. MT624747 | .....  ..  . |
| CRF116_0108. CN. 16. 16YN253. MT624749 | .....  ..  . |
| CRF117_0107. CN. 17. 17ZJ075. MK397789 | K...E  VQ  S |
| CRF118_BC. CN. 12. DH33. KF250409 | .....  ..  . |
| CRF118_BC. CN. 17. YN287_168. MZ063029 | .....  ..  . |

**FIG 2** Amino acid characteristics at the 5 special sites of RT coding region of various HIV-1 subtypes and CRFs. The same amino acid patterns to the K₂₈E₃₂ variant are highlighted by plum purple shadows, and any sequences sharing 1 to 4 same residue to the K₂₈E₃₂ variant are highlighted by light pink shadows. Dot, identity with the topmost sequence.

To test whether the 5 mutations are specific for CRF07_BC, we analyzed the amino acid characteristics at the 5 sites of the RT coding region of other HIV-1 subtypes and CRFs. The results showed that the representative strains of most analyzed subtypes and CRFs do not carry any one of the 5 specific mutations, except the K₂₈E₃₂ variant, as well as several CRF07_BC-involved recombinants (e.g., CRF102_0107, CRF117_0107) that carry 1 to 5 of the specific mutations and might originate via second-generation recombination between the CRF07_BC K₂₈E₃₂ variant and CRF01_AE (Fig. 2). The prevalence of CRF07_BC appeared to be mainly restricted in China and surrounding countries/areas (e.g., the China-Myanmar border area) (20). We further investigated whether these mutations also arose in other regions of the world. HIV-1 subtypes A to D and CRFs 01_AE and 02_AG were the most widely prevalent strains in the world. We analyzed the frequency of these mutations in all available sequences of the 6 subtypes/CRFs. The vast majority of the sequences of the 6 subtypes/CRFs shared the same amino acid feature (63.0%-86.1%) to the CRF07_BC wild-type (WT) strain at the 5 sites of RT coding region, or belonged to the others (12.4%-51.3%) that carried 1 to 4 of the 5 specific mutations and/or other mutations (Table S1). Importantly, no sequences were found to carry the same amino acid feature at the 5 sites to the K₂₈E₃₂ variant (Table S1).

**Evolutionary origin of the K₂₈E₃₂ variant of CRF07_BC.** To trace the origin and evolutionary history of the K₂₈E₃₂ variant, Bayesian phylogenetic analysis was performed.

The origin time of CRF07_BC was estimated to be 1993.6 (95% confidence interval [CI]: 1991.1–1995.4), very close to the earlier estimates (9). In the maximum clade credibility (MCC) tree (Fig. S2), as well as the maximum likelihood (ML) tree (Fig. 3a), all the $K_{28}E_{32}$ variants form a large independent clade that is located at the tip of the tree. The time to the most recent common ancestor (tMRCA) of the $K_{28}E_{32}$ variants was estimated to be 2003.0 (95% CI: 2001.2–2004.4) (Fig. S2), indicating that the variant was formed since 2003. The earliest circulating $K_{28}E_{32}$ variant was detected in 2006, about 3 years later since its origination.

One sequence (green branch in the MCC tree) carrying mutations E28K, K32E, E248V and K249Q was identified to link the $K_{28}E_{32}$ variant clade with the wild-type strains, suggesting that it was an evolutionary intermediate from the WT strain to the $K_{28}E_{32}$ variant. The intermediate was isolated from a man who had sex with men in 2010 and featured by K-E-VQ-T at 28, 32, 248/249, and 338 sites of the RT coding region, respectively (Fig. 3a). The tMACR of the intermediate and the $K_{28}E_{32}$ variants was estimated to be 2000.8 (95% CI: 1998.3-2002.9), and the divergence time of the intermediate from the WT strains was estimated to be 1998.3 (95% CI: 1996.2-2000.2) (Fig. S2). These suggest that the origin of the $K_{28}E_{32}$ variant experienced at least 2 evolutionary steps and in the evolutionary events, 4 mutations E28K, K32E, E248V, and K249Q were first fixed during 1998 to 2000, and then T338S was fixed during 2000 to 2003.

Apart from the $K_{28}E_{32}$ variant and the wild-type, there are several variants carried 1 to 4 mutations at the 5 specific sites. Three variants carrying any 1 or 2 of mutations E28K and K32E were found in the clade of the wild-type strains (Fig. 3a). Interestingly, in the $K_{28}E_{32}$ variant clade, 6 variants carrying one back mutation at site 248 or 249 (V248E and Q249K), and 2 variants carrying one other mutation (V248A or Q249H) were found (Fig. 3a). These results suggest on-going evolution of CRF07_BC regardless of the $K_{28}E_{32}$ variant or the wild-type strains.

Given that the 5 specific residues represent <0.67% (0.11%, 5/440) of analyzed *RT* coding sequence, we investigated whether they alone influence the phylogeny of CRF07_BC. We removed the 5 residues from the RT coding sequences, and re-constructed the ML tree of CRF07_BC. The removing of the 5 sites did not substantially change the tree topology, except the evolutionary intermediate that shows different phylogenetic locations in both ML trees (Fig. 3b). When the 5 sites were removed, the topological location of the intermediate was shifted from a position between the $K_{28}E_{32}$ variant clade and the wild-type clade to a position within the wild-type strain clade (Fig. 3). These results indicate that the 5 mutations are a critical determinant for the evolutionary origin of the $K_{28}E_{32}$ variant. We then investigated whether the 5 specific residues were under positive selection. No one of the 5 residues was identified under significantly positive selection (Table 1), indicating that the generation and expansion of the $K_{28}E_{32}$ variant are less likely a result of positive selection.

We further simulated the expansion dynamics of both the wild-type strain and the $K_{28}E_{32}$ variant of CRF07_BC using Bayesian skyline plot analysis. The wild-type strain experienced a continuous expansion since its origin in early 1990s, and peaked in about 2005, 2 years after the generation of the $K_{28}E_{32}$ variant (Fig. 4). The $K_{28}E_{32}$ variant experienced a growing expansion since its origin in about 2003 (Fig. 4). Accompanied with a continuous decline of the wild-type strains, the $K_{28}E_{32}$ variant was estimated to exceed the wild-type strain in about 2015 in population size.

**Prevalence and distribution of the $K_{28}E_{32}$ variant of CRF07_BC among high-risk groups.** Because systematic HIV-1 molecular epidemiological investigations were previously conducted among MSM in Shenzhen, China, from before 2007 to 2020 (11, 21), and all previously reported *pol* sequences are available in GenBank, we used the sequence data from Shenzhen to analyze the distribution of the $K_{28}E_{32}$ variant, the wild-type strain, as well as others variants (intermediates) of CRF07_BC among different high-risk groups. We found that the $K_{28}E_{32}$ variant mainly appeared among MSM (73.2%), whereas the wild-type strain was mainly prevalent among IDUs (84.8%) (Fig. 5a). It is not surprising that the vast majority of the wild-type CRF07_BC were from

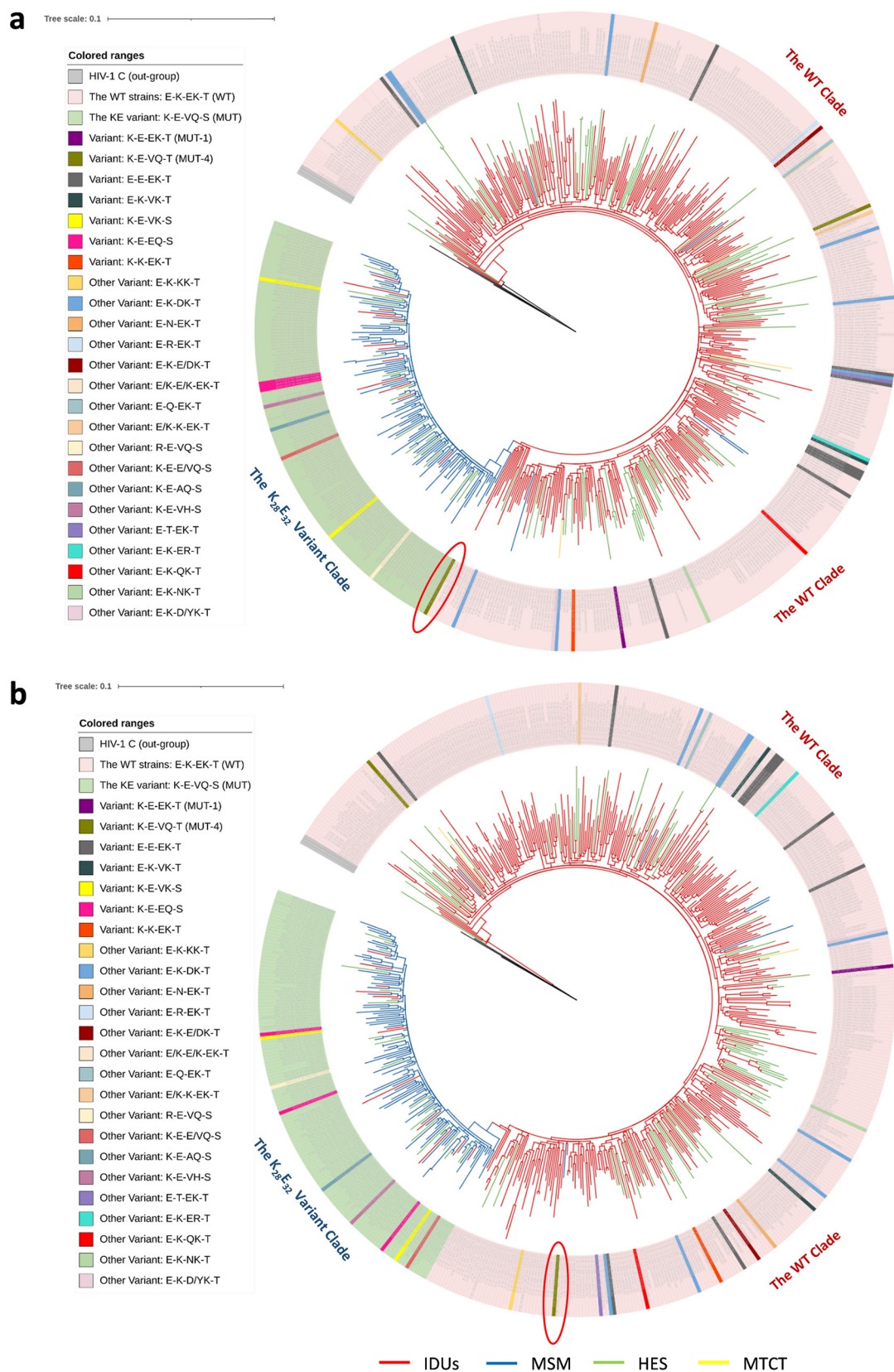

**FIG 3** The ML trees of RT coding sequences of HIV-1 CRF07_BC with (a) and without (b) 5 special sites (28, 32, 248, 248 and 338). A total of 570 HIV-1 CRF07_BC pol sequences were included in the trees and three HIV-1 subtype C strains were used as the out-group. The clades of the $K_{28}E_{32}$ variant and wild-type of CRF07_BC are labeled. The risk groups are highlighted by colored branches, and the $K_{28}E_{32}$ variant, WT, and various intermediates of CRF07_BC are highlighted by colored shadows. A red circle was used to highlight the evolutionary intermediate.

**TABLE 1** Positive selection analysis of RT coding region of CRF07_BC

| Methods | dN/dS[a] | Positively selected sites (PSS) | |
|---|---|---|---|
| | | Sites | no. |
| SLAC[b] | 0.187 | 6, 8, 36, 39, 48, 102, 121, 135, 166, 200, 211, 286, 311, 313, 317, 334, 435, 437 | 18 |
| MEME[c] | 0.175 | 6, 36, 39, 48, 102, 111, 121, 135, 162, 174, 188, 197, 200, 207, 211, 245, 251, 261, 276, 286, 297, 311, 312, 317, 334, 345, 346, 357, 369, 376, 377, 435, 437, 439 | 34 |
| FEL[d] | NA[e] | 6, 36, 39, 102, 121, 135, 200, 211, 245, 286, 311, 317, 334, 435, 437 | 15 |
| FUBAR[f] | NA | 6, 36, 39, 102, 121, 135, 200, 211, 286, 313, 317, 334, 435, 437 | 14 |

[a]This dN/dS represents the ratio of the number of nonsynonymous variants per non-synonymous site (dN) to the number of synonymous variants per synonymous site (dS). The dN/dS values of >1, = 1 and <1 indicate positive selection, neutral evolution and negative (purifying) selection, respectively.
[b]SLAC, single-likelihood ancestor counting.
[c]MEME, mixed effects model of evolution.
[d]FEL, fixed effects likelihood.
[e]NA, not available.
[f]FUBAR, Fast, unconstrained Bayesian approximation.

IDUs since CRF07_BC initially originated among IDUs in early 1990s. However, the proportion of the $K_{28}E_{32}$ variant was significantly higher among MSM (87.4%) than IDUs (3.7%) and the heterosexuals (23.8%) ($P < 0.0001$ for both). Among the heterosexuals, the $K_{28}E_{32}$ variant and wild-type strain accounted for 23.8%, and 61.9%, respectively (Fig. 5a). These results suggest that the $K_{28}E_{32}$ variant was closely associated with homosexual transmission.

We next investigated the dynamics of the $K_{28}E_{32}$ variant and the wild-type strain among IDUs, MSM, and the heterosexuals during the past decades using all available sequences (Fig. 5b to d). The proportion of the $K_{28}E_{32}$ variant appeared to rapidly increase accompanied with a decrease of the wild-type strains before 2010, and remained relatively stable since 2011 among both IDUs and heterosexuals (Fig. 5b and c). However, the proportion of the $K_{28}E_{32}$ variant appeared to slowly decrease from 100% (only one sequence) before 2007 to 70.4% in 2019–2020 among MSM (Fig. 5d).

**The $K_{28}E_{32}$ variant of CRF07_BC significantly improved *in vitro* HIV-1 replication ability.** The crystal structure shows the RT enzyme of HIV-1, like a human right hand, contains 4 subdomains: fingers (1–85 and 118–155), palm (86–117 and 156–236), thumb (positions 237–318), and connection (319–426) (22). Mutations E28K and K32E are located in the finger domain, E248V and K249Q in the thumb domain and T338S in the connection domain (Fig. 6). Although structural simulation suggests that the 5 mutations do not significantly change the RT structure, the significant change of amino acid properties at 28 and 32, as well as 248 and 249 sites might influence the function of RT enzyme. In particular, the residue at site 28 changed from an acidic (Glu) to a basic (Lys) amino acid, while inversely the residue at site 32 changed from a basic (Lys) to an acidic (Glu) amino acid. Furthermore, these mutation sites are not directly located at the RNA/DNA binding domain formed with fingers, palm, and connection, suggest-

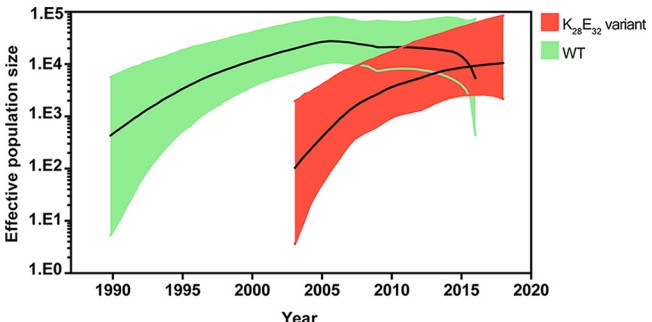

**FIG 4** Population expansion dynamics of the WT and the $K_{28}E_{32}$ variant of CRF07_BC. The solid line and shaded region represent median and 95% HPD (highest probability density) intervals of the effective population size through year. The population dynamics of the $K_{28}E_{32}$ variant (red) and the WT (green) were inferred using the Gaussian Markov Random Field (GMRF) model.

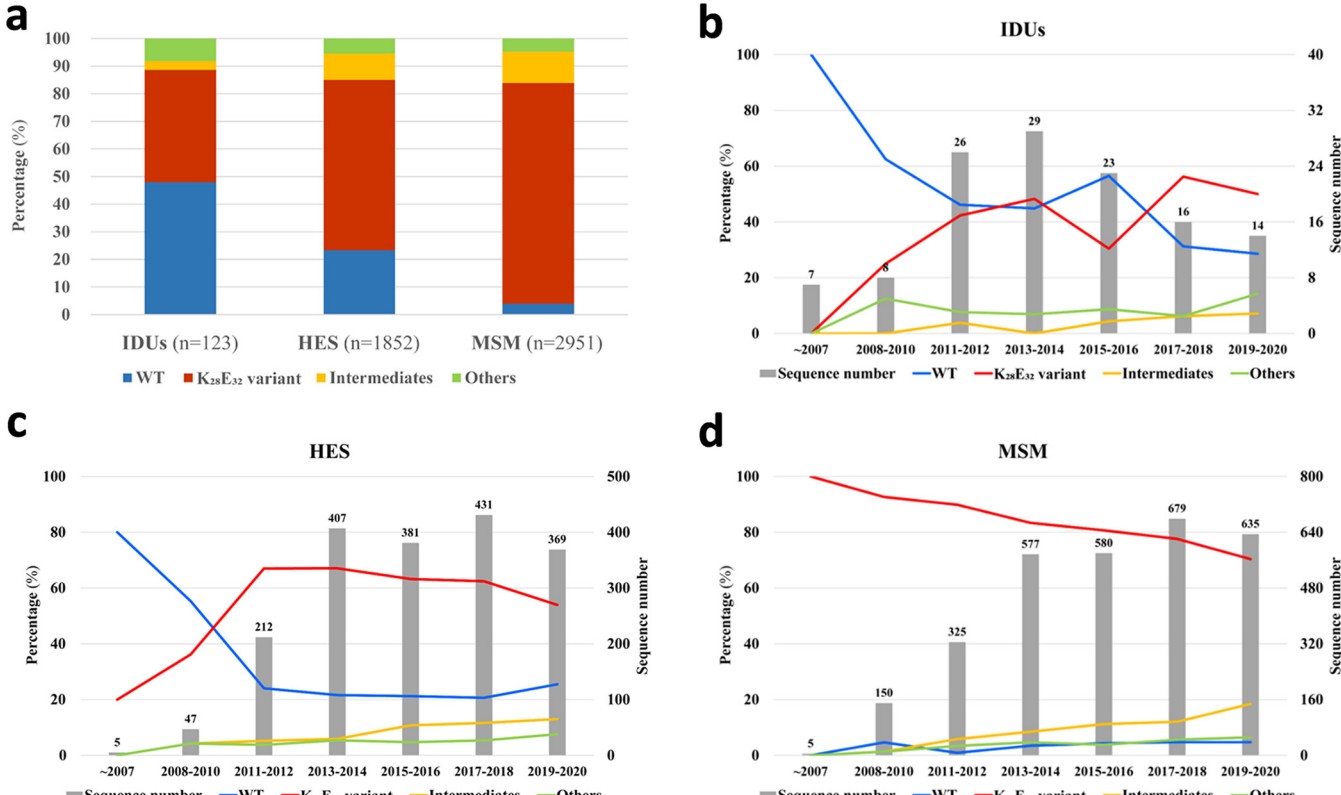

**FIG 5** Prevalence of various CRF07_BC strains (WT, the $K_{28}E_{32}$ variant, intermediates, and others) among different risk groups in Shenzhen city. (a) Comparison of the prevalence of various CRF07_BC strains between 3 major risk groups (IDUs, heterosexuals and MSM). Evolution of the distribution of various CRF07_BC strains among IDUs (b), heterosexuals (c) and MSM (d) from before 2007 to 2020.

ing that functional change of the RT of the $K_{28}E_{32}$ variant might not be involved in the binding of HIV-1 genomic RNA (gRNA).

To determine the influence of the $K_{28}E_{32}$ variant on HIV-1 replication, we constructed the infectious clones of the $K_{28}E_{32}$ variant (NL4-3_07RT-$K_{28}E_{32}$) and the wild-type strain (NL4-3_07RT-WT) by incorporating their RT coding fragments into the full-length HIV-1 NL4-3 molecular clone. Infectious virions were generated in HEK293T cells by transfection. Normalized amounts (10 ng of p24) of the $K_{28}E_{32}$ variant and the wild-type virions were used to infect MT-2 cells. Viral replication was monitored over a period of 12 days by quantifying p24 and viral RNA copies in the culture supernatant. Both NL4-3_07RT-$K_{28}E_{32}$ and NL4-3_07RT-WT showed consistent replication dynamics. HIV-1 RNA and p24 levels continuously increased, especially during 6–10 days after infection (Fig. 7a). HIV-1 RNA level peaked at day 10, while the p24 level still slowly increased to day 12, regardless the variant and the wild-type strain, suggesting that the p24 level might be slightly delayed to viral RNA level. Since day 8, NL4-3_07RT-$K_{28}E_{32}$ generated significantly higher HIV-1 RNA and p24 levels than NL4-3_07RT-WT ($P < 0.01$), suggesting that the $K_{28}E_{32}$ variant has greater *in vitro* replication capacity than the wild-type strain.

The $K_{28}E_{32}$ variant has 5 specific mutations. To investigate the crucial mutations influencing HIV-1 replication, we further constructed 6 additional mutants (Table 2), and measured their replication dynamics in MT-2 cells (Fig. 7a). All 6 mutants had consistent replication dynamics with the wild-type strain and the $K_{28}E_{32}$ variant, as reflected by HIV-1 RNA and p24 levels. Among the 6 mutants, MUT-1 appeared to have the greatest replication capacity, followed by MUT-2/MUT-4, and MUT-5 (Fig. 7b). In particular, the replication capacity of the MUT-1 was similar but slightly greater than the $K_{28}E_{32}$ variant. Compared to the wild-type strain, both MUT-1 and the $K_{28}E_{32}$ variant share common mutations E28K and K32E, indicating that the 2 mutations mainly contribute to the improvement of HIV-1 RT replication capacity *in vitro*. The MUT-3 and

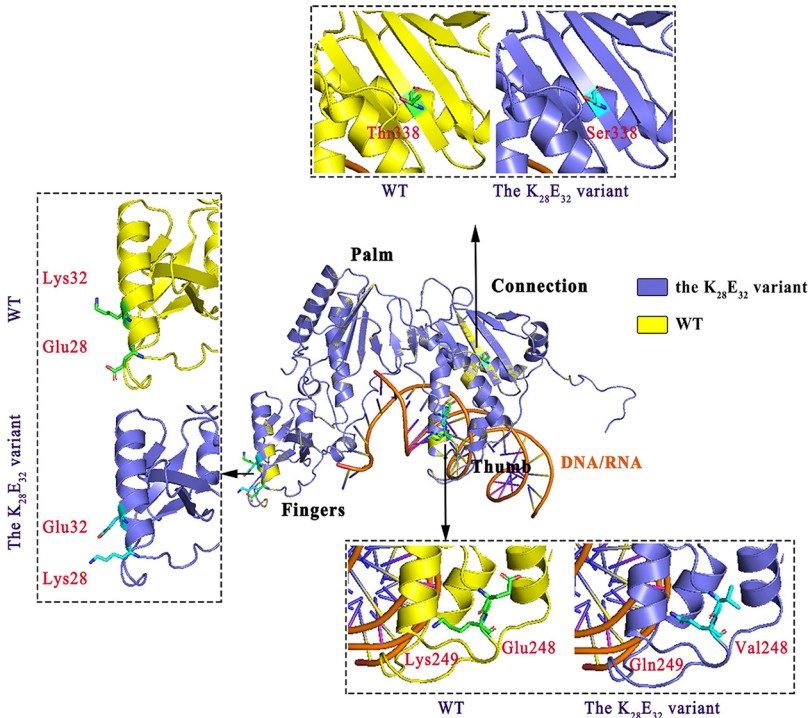

**FIG 6** Structural comparison between the CRF07_BC wild-type (yellow) and the $K_{28}E_{32}$ variant (blue). The original side chains are marked by green, while the mutated side chains are marked by light blue.

MUT-6 carried different amino acids at sites 248 and 249, but had similar lower replication capacity among the 6 mutants, suggesting that mutations E248V and K249Q might have relatively less influence on the RT replication capacity. Compared to the wild-type strain, both MUT-3 and MUT-6 shared mutation T338S, and had similar slightly lower replication activity than the wild-type strain (Fig. 7b), suggesting that the T338S might also have less influence on the RT replication ability. In addition, the MUT-5 carries E28K, K32E and T338S and had similar replication capacity to the wild-type strain. The possible reason might be the improvement of replication capacity by E28K and K32E was counteracted by the T338S that reduces the RT activity.

**The $K_{28}E_{32}$ variant of CRF07_BC significantly improved early and late reverse transcription and nuclear localization.** We further determined the effect of various RT mutants of CRF07_BC on minus strand strong-stop (early RT) and second-strand transfer (late RT). The early and late RT products of the wild-type strain peaked at 2 and 4 h postinfection, respectively, and then slowly decreased (Fig. 8a to d). Compared to the wild-type strain, the $K_{28}E_{32}$ variant, MUT-1 and MUT-4 all showed substantially more early and late RT products at each time point. The $K_{28}E_{32}$ variant showed 2.33–2.69-fold improvement in early and late RT products compared to the wild-type strain. In particular, MUT-1 showed the greatest improvement in both early and late RT products, and its early and late RT products at 4 h postinfection were 2.93–3.63 and 1.26–1.35-fold higher than the wild-type strain and the $K_{28}E_{32}$ variant, respectively (Fig. 8a to d). Comparison of 5 specific amino acids among the $K_{28}E_{32}$ variant, MUT-1, and the wild-type strain suggest that mutations E28K and K32E play crucial role in the improvement of early and late reverse transcription.

We also examined the ability of various RT mutants for nuclear localization by quantifying 2LTR circle formation. 2LTR products continuously increase up to 48 h postinfection for WT and all mutants (Fig. 8e). Analysis of the 2LTR products at 48 h postinfection showed that the $K_{28}E_{32}$ variant, MUT-4 and MUT-1 had significantly higher 2LTR products than the wild-type strain; while in contrast, MUT-3 exhibited substantially lower amount of 2LTR products. Higher level of 2LTR products by the $K_{28}E_{32}$ variant may be simply attributed to

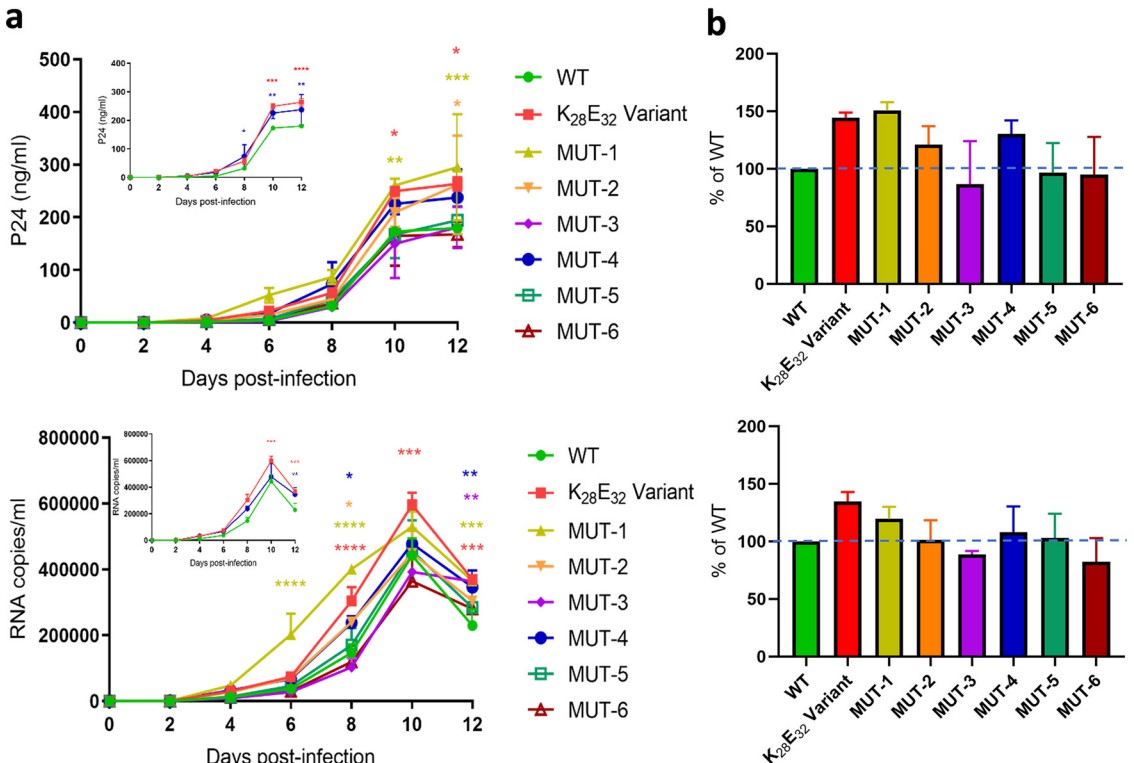

**FIG 7** Replication dynamic of the wild-type, $K_{28}E_{32}$ variant, and six mutants of HIV-1 CRF07_BC. (a) Measurement of p24 and viral RNA in the culture supernatant by ELISA and RT-qPCR, respectively. For clarity, the comparisons of the $K_{28}E_{32}$ variant and the evolutionary intermediate (MUT-4) to the WT strain are individually displayed in small panels. (b) Relative effect of the $K_{28}E_{32}$ variant and six mutants of HIV-1 CRF07_BC to the WT strain of HIV-1 CRF07_BC at day 10 postinfection. The P24 and RNA levels of the WT strain were defined as 100% and highlighted by a dotted line. Statistical analysis was performed by 2-way ANOVA (multiple comparison). *, $P < 0.05$; **, $P < 0.01$; ***, $P < 0.001$.

higher accumulation of late RT products that enhances the pre-viral DNA nuclear translocation.

## DISCUSSION

HIV-1 is one of the most variable RNA viruses with high mutation rate and recombination potential caused by the error-prone nature and the template-jump mechanism of RT enzyme in HIV-1 replication, respectively (23–25). High mutation rate and recombination capacity of HIV-1 are related with its survival by maintaining the balance between transmissibility and virulence (infectiousness-virulence tradeoff) under the action of natural selection (26–28). Most HIV-1 mutations are neutral and/or deleterious, and only a small proportion of mutations are beneficial (23). The beneficial mutations are often associated with drug resistance to various antiretroviral agents (29), or immune escape from existing

**TABLE 2** Specific amino acids of the wild-type, $K_{28}E_{32}$ variant, and six mutants of CRF07_BC

| CRF07_BC strains | Amino acids at sites of RT | | | |
| --- | --- | --- | --- | --- |
| | 28 and 32 | 248 and 249 | 338 | Pattern |
| Wild-type | EK | EK | T | E-K-EK-T |
| $K_{28}E_{32}$ variant | KE | VQ | S | K-E-VQ-S |
| MUT-1 | KE | EK | T | K-E-EK-T |
| MUT-2 | EK | VQ | T | E-K-VQ-T |
| MUT-3 | EK | EK | S | E-K-EK-S |
| MUT-4 | KE | VQ | T | K-E-VQ-T |
| MUT-5 | KE | EK | S | K-E-EK-S |
| MUT-6 | EK | VQ | S | E-K-VQ-S |

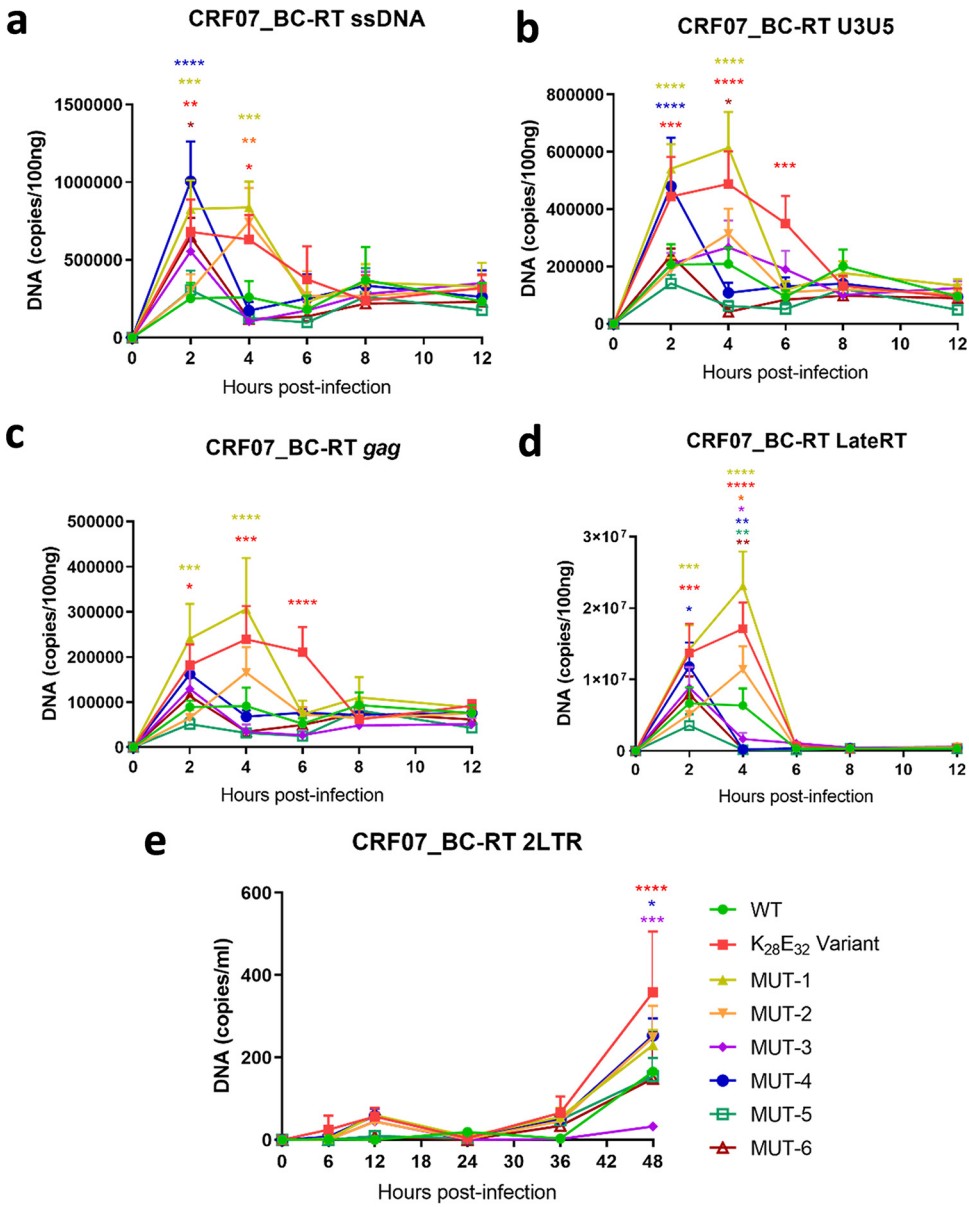

**FIG 8** Replication dynamic of the wild-type, $K_{28}E_{32}$ variant, and six mutants of HIV-1 CRF07_BC. (a to e) ssDNA, *U3U5*, *gag*, late RT, and 2LTR products, respectively. Statistical analysis was performed by 2-way ANOVA (multiple comparison). *, $P < 0.05$; **, $P < 0.01$; ***, $P < 0.001$.

neutralizing antibodies and/or cytotoxic T lymphocyte (CTL) response (30, 31). Recently, a highly virulent variant of subtype B HIV-1 was identified in the Netherlands, and the variant was associated with higher viral load and rapid loss of CD4 T cells (32). In this study, we identified a new variant of CRF07_BC HIV-1 that shows higher *in vitro* replication ability and is mainly circulating among MSM.

HIV-1 exists in quasispecies with one or more mutations in host, and only one or few HIV-1 founder (or fitness) variants can be effectively transmitted from one host to another under strong transmission bottleneck (33, 34). In evolution, HIV-1 strains from both the donors and the recipients are closely genetically related. By tracing the genetic relatedness and identity, HIV-1 transmission link among infected individuals can be identified at local and global scales (17, 18). The fitness variants with increased transmissibility and/or decreased virulence could have higher potential to spread and form large transmission networks, such as the observations in SARS-CoV-2, where the newly emerging Omicron variant with higher transmissibility but relatively lower

virulence is replacing the earlier highly virulent Delta variants (35). Transmission network analysis can effectively identify high-risk HIV-1 transmission networks (groups) and was previously used as an important tool to guide precision intervention for effective HIV/AIDS control (19, 36, 37). Using transmission cluster analysis, previous studies identified some distinct phylogenetic (transmission) clusters of circulating HIV-1 subtypes and CRFs (7, 38–40). Although no cluster-specific amino acid patterns were identified, some HIV-1 CRF01_AE clusters appeared to have stronger virulence and were associated with lower CD4-T cell count and/or higher viral load (41, 42). It's worth noting that the highly virulent clusters were rarely associated with improved replication ability of HIV-1 RT enzyme (32). In this study, using transmission network analysis, we identified a new $K_{28}E_{32}$ variant of HIV-1 CRF07_BC that has higher *in vitro* replication ability than the wild type. The finding and identification of the $K_{28}E_{32}$ variant suggest that transmission network analysis can also be used as a robust tool to find and identify newly emerging highly adapted variants. Given substantial effects in reducing HIV-1 transmission among high-risk groups such as MSM, transmission network analysis has been incorporated into the national guidelines for the routine monitoring and intervention of HIV-1 transmission by China CDC since 2021. National implementation of transmission network monitoring will benefit the finding of newly emerging HIV-1 variants overrepresented in transmission clusters in the future.

The co-circulation of multiplex subtypes inevitably resulted in the on-going generation of various inter-subtype recombinants (4, 24, 43), increasing HIV-1 genetic diversity and exacerbating the epidemic in the developing world. Currently, at least 120 CRFs have been identified globally (https://www.hiv.lanl.gov/content/sequence/HIV/CRFs/crfs.comp) (24). The vast majority of the CRFs was only associated with sporadic infections, and only few CRFs caused regional epidemics. Currently, 4 HIV-1 CRFs (i.e., CRF01_AE, CRF07_BC, CRF08_BC and CRF55_01B) had resulted in large-scale epidemics (>10% for each) in China, and CRF01_AE and CRF07_BC are becoming the most predominant HIV-1 strains (5, 6). CRF01_AE was mainly circulating among heterosexuals and MSM at early HIV-1 epidemic, and remained the most predominant HIV-1 strain in MSM until the past few years (11, 40). CRF07_BC, CRF08_BC and CRF55_01B originated in China and were mainly restricted to China (8, 44, 45). In particular, CRF07_BC and CRF08_BC were 2 sister CRFs that originated among IDUs in Yunnan in a narrow time window (1990-1993), but experienced different spread and expansion history (9, 46, 47). The prevalence of CRF08_BC was mainly restricted to heterosexuals and IDUs in limited regions, while CRF07_BC spread from IDUs to heterosexuals, and further to MSM. In particular, CRF07_BC experienced a very rapid expansion among MSM since 2006 and was replacing CRF01_AE to be the most predominate HIV-1 subtype among MSM (11, 13). Accompanied with the growing HIV-1 prevalence among MSM, CRF07_BC is expected to eventually be the most predominate HIV-1 strains in China, regardless of IDUs and sexual high-risk groups. However, the reason for the rapid expansion of CRF07_BC remains unknown.

The newly identified $K_{28}E_{32}$ variant of CRF07_BC accounted for a significantly higher proportion among MSM than IDUs ($P < 0.01$), and may be responsible for the rapid expansion of CRF07_BC among MSM (11, 13, 48). First, the $K_{28}E_{32}$ variant originated among MSM in about 2003, earlier than the growing expansion of CRF07_BC among MSM (11, 49). Second, the $K_{28}E_{32}$ variant carried 5 specific mutations in RT coding region, which confers its high *in vitro* replication capacity to generate more virions than the wild type. Third, the $K_{28}E_{32}$ variant was overrepresented in large transmission networks among MSM, suggesting that it is genetically relatively conserved and can effectively break the mucosal transmission bottleneck to spread among MSM. Interestingly, 2 recent studies divided CRF07_BC into 2 clusters, CRF07_BC_O and CRF07_BC_N, and demonstrated that CRF07_BC_N was mainly circulating and was more transmissible among MSM than CRF07_BC_O (49, 50). According to the phylogeny and epidemiological trait, CRF07_BC_N was highly suspected to be the $K_{28}E_{32}$ variant.

The evolution of the $K_{28}E_{32}$ variant experienced at least 2 stages, from the wild-type to an intermediate (KP178444, MUT-4), and from the intermediate to the $K_{28}E_{32}$ variant. Of 5 specific mutations in the $K_{28}E_{32}$ variant, mutations E28K and K32E play a crucial role in enhancing *in vitro* replication capacity of the RT enzyme. The intermediate (MUT-4) had accumulated 4 of the 5 specific mutations, except T338S, and exhibited slightly lower level of *in vitro* HIV-1 replication than the $K_{28}E_{32}$ variant, but significantly higher level than the wild type. Because the full-length genomic sequence of the intermediate is not available, any differences in other genes between the intermediate and the $K_{28}E_{32}$ variant remains unclear. The appearance of the $K_{28}E_{32}$ variant to replace the evolutionary intermediate and the wild-type among MSM might be simply attributed to its stronger replication ability, a founder effect and/or the accumulation of additional adaptive mutations in other genes (33, 51). On the other hand, we detected back mutations (V248E or Q249K) and new mutations (V248A or Q249H) at 248 or 249 sites in several variants. The appearance of back and new mutations at 248 and 249 sites not only supports less influence of amino acids at 248 and/or 249 sites of RT coding region on *in vitro* HIV-1 replication, but also indicates an on-going evolution and adaption of the $K_{28}E_{32}$ variant to MSM and even other high-risk groups. Furthermore, other variants carrying any 1 or 2 mutations E28K and K32E were found in the clade of wild-type strains (Fig. 3a). In particular, 1 variant carrying both E28K and K32E (MUT-1) might have a higher level of *in vitro* HIV-1 replication capacity than the $K_{28}E_{32}$ variant, other mutants, and the wild type. The potential risk of this variant evolving to a new $K_{28}E_{32}$ variant-like variant among IDUs should be highly watched.

The speed–fidelity trade-off determines the mutation rate and virulence of an RNA virus, and the extremely high mutation rate of HIV-1 is a consequence of error-prone replication of the RT enzyme (23, 26). It is interesting that CRF07_BC exhibits higher transmission advantage than other HIV-1 subtypes (e.g., CRF01_AE and B) circulating in China (14), but had significantly lower average genetic distance than the latter (11). Because the $K_{28}E_{32}$ variant was associated with rapidly growing transmission networks among MSM, it was not surprising that the $K_{28}E_{32}$ variant had significantly lower genetic distances (mean distance: 0.021) than the wild-type (0.033) ($P < 0.0001$, *t* test). The evolutionary rate of the $K_{28}E_{32}$ variant was estimated to be $1.781 \times 10^{-3}$, also substantially lower than that ($3.945 \times 10^{-3}$) of the wild type. The 5 mutations in the RT coding region are specific features to define the $K_{28}E_{32}$ variant, and are involved in its origin; however, they were not subject to significantly positive selection. In view of the *in vitro* replication advantage conferred by the 5 specific mutations, selective sweep might contribute to the stability of the 5 specific residues and the lower genetic diversity of the $K_{28}E_{32}$ variant (52). On the other hand, we did not determine the RT fidelity of the $K_{28}E_{32}$ variant in this study; therefore, it is unclear whether increased *in vitro* replication ability, but decreased mutation rate of the $K_{28}E_{32}$ variant, are involved in the fidelity of the RT enzyme, and if so, which mutations may affect and/or determine the replication fidelity of the RT enzyme. Some previous studies reported that some HIV-1 CRF01_AE transmission clusters were associated with rapid loss of CD4 T cell counts and/or higher viral load, implying an association of these variants with rapid disease progression (41, 42, 53). Compared to CRF01_AE and CRF55_01B, CRF07_BC, they appeared to be associated with relatively lower viral load and higher CD4 T cell count among MSM, and might have a relatively slow disease progression (48, 54). This difference might be involved in the fact that almost all CRF07_BC strains belong to R5 (CCR5 tropism) virus, while the majority of CRF01_AE strains were X4 (CXCR4 tropism) virus (53, 55). We further investigated the tropism of the $K_{28}E_{32}$ variant and the wild-type using all available full or near full-length CRF07_BC ($n = 44$) sequences from the HIV database. There were 8 $K_{28}E_{32}$ variants and 36 wild-type strains. All these sequences, regardless of the $K_{28}E_{32}$ variant and the wild-type, were predicted to have CCR5 tropism using geno2pheno and the R5-X4 pred tool (56, 57). This suggests that co-receptor tropism does not contribute to the rapid spread and adaption of the $K_{28}E_{32}$ variant among MSM.

There are 2 limitations of this study. First, although we demonstrated that the $K_{28}E_{32}$ variant have a stronger *in vitro* replication ability, we did not investigate whether this new CRF07_BC variant affects and/or changes disease progress since the used sequences were mainly download from the HIV database and the related clinical information are unavailable. Second, apart from the 5 specific mutations in the RT coding region, other genes (e.g., Vif, Nef and Tat) of the $K_{28}E_{32}$ variant also carried specific mutations (data not shown). HIV-1 accessory proteins not only play crucial roles in HIV-1 replication, assembly, and survival, but also counteract host restriction factors (e.g., APOBEC3G and Tetherin) (58, 59). Whether the specific mutations in the accessory genes of the $K_{28}E_{32}$ variant affect HIV-1 life cycle and/or their activities to escape host immunity by counteracting cellular restriction factors deserves further investigation in future.

Taken together, using transmission network analysis, we identified and characterized a new CRF07_BC $K_{28}E_{32}$ variant that carries 5 specific mutations in the RT coding region, and exhibits higher *in vitro* HIV-1 replication ability than the wild type. Extremely high prevalence of the $K_{28}E_{32}$ variant among MSM and its overrepresentation in large transmission clusters suggest its association with the rapid expansion of CRF07_BC among MSM in recent years. The emergence and subsequent predominance of the $K_{28}E_{32}$ variant among MSM could be ascribed to its higher *in vitro* replication ability and/or simply a founder effect of this variant being propagated among groups that are currently being infected in China (7, 51). This could allow public health officials to use this marker (5 specific mutations) to target prevention measures, like aggressive treatment provision to MSM population and persons infected with this variant (37). It could also be that other viral characteristics linked to the $K_{28}E_{32}$ variant are responsible for the quick spread of this variant within a risk network. Further characterization of this possibility is needed, which may identify ways to interrupt any innate transmission advantage that these viruses have (60).

## MATERIALS AND METHODS

**HIV-1 CRF07_BC *pol* sequence analysis.** CRF07_BC *pol* sequences from 1997–2013 were downloaded from the HIV database (https://www.hiv.lanl.gov/components/sequence/HIV/search/search.html) on December, 2015. After removing those without geographic origin and sampling year, 1195 *pol* sequences (899 nt with a location of 2289–3187 nt in HXB2) were used for transmission cluster and evolutionary analyses. A ML tree was constructed using FastTree version 2.1 (http://meta.microbesonline.org/fasttree/), and HIV-1 evolutionary (or transmission) clusters were identified using ClusterPicker 1.2.1 with parameters of initial threshold: 0.9, main support threshold: 0.9, genetic distance threshold: 4.5 (61). The cluster containing over 20 sequences was defined as a large cluster for further analyses. The sequences that were unable to form an evolutionary cluster were defined as 'non-cluster' sequences. To characterize the features of sequences in clusters versus not in clusters (non-cluster), each amino acid sequence was translated from the RT coding sequence, and the sequence logo was generated using WebLogo Version 2.8.2 (http://weblogo.berkeley.edu/logo.cgi). Significance was evaluated using Viral Epidemiology Signature Pattern Analysis (VESPA: https://www.hiv.lanl.gov/content/sequence/VESPA/vespa.html). Viral tropism was determined using geno2pheno and the R5-X4 pred tool (56, 57).

**Phylogenetic and molecular clock analysis.** A total of 570 p51 (RT coding) sequences of CRF07_BC strains with known demographic information (e.g., sampling date, location, and risk factors) were subjected to phylogenetic reconstruction using approximate maximum likelihood with PhyML 3.0 program. Among them, 207 sequences were from the newly diagnosed HIV-1-positive patients in Yunnan province and Shenzhen city from the year 2010 to 2016 in this study, who were participants in the National Key S&T Special Projects on Major Infectious Diseases. All participants signed written informed consents prior to sample collection, and completed standardized questionnaires that included demographic data. This study was reviewed and approved by the ethics committees of the Beijing Institute of Microbiology and Epidemiology. The other 363 sequences, which were sampled in China from 1997 to 2018, were downloaded from HIV database (http://www.HIV.lanl.gov) in September, 2021.

The GTR + G+I nucleotide substitution model was selected by using Smart Model Selection (SMS) (62). The heuristic tree search was performed using the SPR branch-swapping algorithm, and the branch support was calculated with the approximate likelihood-ratio (aLRT) SH-like test (63, 64). The final maximum likelihood tree was visualized by using the program MEGA v6.06 and iTol v6 (https://itol.embl.de/).

We performed root-to-tip divergence analysis using TempEst v1.5.1 to evaluate the sampling time signal for data (R squared $>0.7$) (65). After removing a few sequences showing incongruent temporal patterns, 527 sequences were subjected to subsequent analysis. Bayesian demographic reconstruction of HIV-1 CRF07_BC was conducted by BEAST v1.10.4 Packages with a GTR+G+I nucleotide substitution model, an uncorrelated lognormal relaxed clock model, a Bayesian Skyline tree prior, $5 \times 10^8$ length of

chain sampling frequency of 1000 (66). All phylogenetic trees were visualized by Figtree v1.4.2 and MEGA v6.06. To explore population growth, 395 and 131 RT coding sequences from CRF07_BC wild-type and the $K_{28}E_{32}$ variant was subject to Bayesian skyline plot analysis implemented in BEAST v.1.10.4 Packages.

**Natural selection analysis.** Site-specific detection methods implemented in Datamonkey (http://datamonkey.org), including single-likelihood ancestor counting (SLAC), mixed effects model of evolution (MEME), fixed effects likelihood (FEL), and fast, unconstrained Bayesian approximation (FUBAR), were used to identify positively selected sites in the RT coding region of CRF07_BC (67). Codon positions with a $P$-value $< 0.05$ for the SLAC, FEL, or MEME model, or with a posterior probability $>0.95$ for the FUBAR method, were considered to be under significantly positive selection.

**Cell culture.** HEK293T cells and TZM-bl cells were cultured in DMEM medium (Gibco) containing 10% fetal bovine serum (FBS) (Gibco) and 100 units/mL penicillin and 100 $\mu$g/mL streptomycin. MT-2 cells were cultured in RPMI 1640 medium (Gibco) containing 10% FBS and 100 units/mL penicillin and 100 $\mu$g/mL streptomycin.

**Construction of infectious clones of the wild-type, $K_{28}E_{32}$ variant, and other related mutants of HIV-1 CRF07_BC.** In order to obtain a recombinant CRF07_BC infectious clone, the RT coding region (HXB2:2550-3870) of HIV-1 subtype B infectious clone pNL4-3 was replaced by a RT coding fragment from a CRF07_BC strain (Accession Number: HQ215552) (Fig. S3). The infectious clone plasmid was linearized by restriction endonuclease digestion and purified by Apal/EcoRI extraction. To generate the infectious clones of various CRF07_BC RT variants (mutants), the Q5 site-directed mutagenesis kit (NEB) was used to introduce corresponding substitutions into the recombinant CRF07_BC infectious clone. The substitution sites were confirmed by PCR and Sanger sequencing. A total of 7 CRF07_BC RT variants were constructed, including the $K_{28}E_{32}$ variant (MUT), and 6 related mutants (MUT-1-MUT-6). The characteristic amino acid sites of the CRF07_BC wild-type (WT) strain, $K_{28}E_{32}$ variant (MUT), and 6 related mutants (MUT-1-MUT-6) are listed in Table 2.

**HIV-1 CRF07_BC stocks.** The recombinant CRF07_BC plasmids were transfected into HEK293T cells using Lipofectamine 2000 reagent (Thermo Fisher Scientific) to generate virus stock. Culture supernatants were collected at 48 to 72h posttransfection. Infectious virions were detected by tissue culture infectious dose 50 (TCID50). HIV-1 p24 antigen expression was detected by enzyme-linked immune sorbent assay (ELISA). The correction of the mutations in generated mutant virions was further confirmed by RT-PCR and Sanger DNA sequencing using the RNA from the supernatant.

***In Vitro* replication capacity of the wild-type, $K_{28}E_{32}$ variant, and 6 mutants of HIV-1 CRF07_BC.** To determine the replication kinetic of various CRF07_BC RT variants, a total of $8 \times 10^5$ MT-2 cells were infected with the viral supernatants containing 10 ng p24 antigen. After 6 h of incubation, the cells were washed twice with PBS, and fresh medium (RPMI 1640 containing 10% FBS) was added to each well. Infected cells were maintained at 37°C with 5% $CO_2$ and the supernatants were collected at the indicated time points of 0, 2, 4, 6, 8, 10, and 12 days after infection. The p24 antigen content in the supernatant was detected by ELISA. Viral RNA was extracted from the supernatant using Viral RNA minikit (QIAamp) and a previously established RT-qPCR assay was performed to determine the mRNA copies in the viral supernatants (68).

**Measurement of HIV-1 replication intermediates.** As described above, MT-2 cells were infected with various CRF07_BC RT variants. The supernatants were collected at time points of 0 h, 2 h, 4 h, 6 h, 8 h, 12 h, 24 h, 36 h, and 48 h after infection. After removing the supernatant, the cells were washed with PBS and collected for extraction of genomic DNA with DNA minikit (QIAamp). HIV-1 replication intermediates (ssDNA, U3U5, Gag, late RT and 2LTR fragments) were measured by qPCR assays as previously described (69). The primers and probes are available in ref (69).

The qPCR assays were performed by using the GoldStar Probe Mixture (CoWin Biosciences). A 15 $\mu$L qPCR system was set up, containing $1 \times$ gold star TaqMan mixture, 0.2 $\mu$M (each) forward and reverse primers, 0.2 $\mu$M probe, and 500 ng template DNA or non-template control (NTC). The reactions were performed using LightCycle 480 (Roche), and the reaction condition was pre-denaturation at 95°C for 10 min, followed by 40 cycles of denaturation at 95°C for 15s, and annealing and extension at 60°C for 1 min.

**Statistical analysis.** All data were analyzed using the GraphPad Prism software. Statistical evaluation was performed by Student's *unpaired t test* or One-Way ANOVA with Tukey's multiple-comparison test. Data are presented as means $\pm$ SD or as described in the corresponding legends.

**Data availability.** The *pol* sequence alignments were available at https://github.com/mayingying1997/CRF_07BC-sequence.git. The sequences obtained in this study were submitted to GenBank and the accession numbers are ON241448-ON241654. Other sequences used in this study were downloaded from GenBank. All the software used in this study are available from open source.

## SUPPLEMENTAL MATERIAL

Supplemental material is available online only.

**SUPPLEMENTAL FILE 1**, PDF file, 0.7 MB.

## ACKNOWLEDGMENTS

We thank Davey Smith at the Division of Infectious Diseases and Global Public Health, University of California San Diego, for his kind help in an earlier version of this paper and Jin Zhao at Shenzhen CDC for her help during the revision of this paper. We also thank Yi-Qun

Kuang at First Affiliated Hospital of Kunming Medical University, Kunming Medical University and Yanpeng Li at Shanghai Public Health Clinical Center, Fudan University for their suggestions on the paper.

This work was supported by the grants from the National Natural Science Foundation of China (32170147, U1302224 and 81601802), and the State Key Laboratory of Pathogen and Biosecurity (AMMS).

The funders had no role in study design, data collection, analysis, or preparation of the article.

C.Z. conceived and designed the study. C.Z. and L.L. supervised this study. J.H. and Y.-H.Z. collected and analyzed the sequence data. J.H., Y.-H.Z. and Y.M. performed the evolutionary analyses. J.H., G.Z., D.Z., and B.Z. performed the experiments. J.H., T.C., and L.W. performed the structural simulation of RT enzyme. C.Z., J.H., Y.-H.Z., J.-H.W. and L.L. interpreted the results. C.Z., J.H., and Y.-H.Z. drafted the paper. J.-H.W. revised the paper. All authors read and approved the final paper.

We declare that there are no conflicts of interest.

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
