## [Reviewer comments · Microbiology Spectrum]

Microbiology Spectrum

A new HIV-1 K28E32-RT variant associated with the rapid expansion of CRF07_BC among men who have sex with men

Jingwan Han, Yan-Heng Zhou, Yingying Ma, Guoxin Zhu, Dong Zhang, Bo Zhu, Tong Cheng, Lanfeng Wang, Jian-Hua Wang, Lin Li, and Chiyu Zhang

Corresponding Author(s): Chiyu Zhang, Shanghai Public Health Clinical Center, Fudan University

Review Timeline:

Submission Date:	July 13, 2022
Editorial Decision:	August 27, 2022
Revision Received:	September 1, 2022
Accepted:	September 15, 2022

Editor: Takamasa Ueno

Reviewer(s): The reviewers have opted to remain anonymous.

Transaction Report:

DOI: <https://doi.org/10.1128/spectrum.02545-22>

August 27, 2022

Prof. Chiyu Zhang
Shanghai Public Health Clinical Center, Fudan University
2901 Cao Lang Road, Jinshan District, Shanghai, China
Shanghai
China

Re: Spectrum02545-22 (A new HIV-1 K₂₈E₃₂-RT variant causes rapid expansion of CRF07_BC among men who have sex with men搜索复制)

Dear Prof. Chiyu Zhang:

I agree with the comments of the new reviewer pointing out that the authors did not adequately address the concerns raised by the previous reviewer. This revision would be a final opportunity with this journal to modify the manuscript in response to the concerns raised by both reviewers.

Link Not Available

Sincerely,

Takamasa Ueno

Journals Department
Reviewer comments:

Reviewer #1 (Public repository details (Required)):

It appears that the paper is compliant with this requirement. Most of the data were downloaded from public repositories but the methods describes genotypes from 207 individuals collected with written informed consent, and a range of 207 GenBank accession numbers are provided.

Reviewer #1 (Comments for the Author):

Han et al combine molecular epidemiology, phylogenetics and molecular biology approaches to characterize, and investigate potential viral genetic features underpinning the rapid spread of a novel K28E32 variant of CRF07_BC in China. The presented analyses are well undertaken. The authors have also addressed many of the comments provided by a previous reviewer, and these changes have improved the manuscript.

However, some of the original reviewer's comments have not been adequately addressed.

Specifically:

1. As the original reviewer correctly pointed out, there are no data in the present paper that show that the K28E32 variant is more transmissible. The authors provide data supporting this variant as having a higher replication capacity and superior RT activity in vitro, but this does not necessarily equate to increased transmissibility between hosts. The original reviewer also correctly emphasized that founder effects, followed by subsequent dissemination of this new variant in MSM as a result of factors that are completely unrelated to viral genetics, cannot be ruled out as the main reason for this variant's spread in this new population. ON the latter point, the original reviewer emphasized that founder virus effects should be emphasized *from the start* (see discussion point #1 in original reviews).

The authors indicates that they have "toned down" their claims, but the manuscript still contains statements claiming that their data support transmissibility, as well as statements that claim that viral genetics are the underlying mechanism for this strain's spread, without adequately acknowledging founder virus effects from the outset. These statements must be revised, in particular the abstract, importance and concluding paragraph of the study. They include:

-> abstract "This variant... more likely formed transmission clusters and drive the rapid expansion of CRF07_BC....". The authors should avoid this causative language, as it is not supported by the data. Instead say, e.g. "the variant.. was overrepresented in transmission clusters, suggesting that it could have driven the rapid expansion of CRF07_BC in MSM, though founder effects cannot be ruled out"

-> importance statement "This study identified five crucial mutations that answers why CRF07_BC had a rapidly increasing prevalence". The authors must remove strong causation claims such as this.

-> lines 178-180: "The generation of the C28E32 variant was mainly involved in the adaption of CRF07_BC to individuals with reported homosexual risk". This strong causative statement is not supported by the data. These types of statements must be toned down and founder effects must be acknowledged as an alternative explanation from the beginning.

-> line 358 "This indicates that co-receptor tropism does not contribute to the rapid spread of K28E32 in MSM". This cannot be concluded from the data. Change to "This suggests"

-> line 370-372 "We identified...a new CRF07_BC variant that.....has a transmission advantage over wild-type". This statement is not supported by the data.

-> lines 372-374 in the final concluding paragraph of the paper state that founder effects could explain their observations, but this statement comes out of the blue with no context, as if it was simply added to address the reviewer concern. As such, it reads like a direct contradiction of the prior sentence (which claims that the virus has a transmission advantage), rather than an explicit acknowledgement that this could represent an alternative mechanism. The final sentence further adds to the confusion as it seems to walk back the transmission statements as being far from conclusive (i.e. "any innate transmission advantage that these viruses have") and in fact the authors concede in their rebuttal letter that future directions include investigating whether this virus is more transmissible

2. The original reviewer pointed out that evidence linking the K38E32 variant to correlates of HIV disease progression would strengthen the study. The revised manuscript (lines 360-369) indicates that this was not possible because "sequences were downloaded from HIV database and related clinical information were unavailable". However the methods section describes the recruitment of 207 participants with HIV with written informed consent who underwent HIV genotyping, for whom sociodemographic (and presumably clinical) data were also collected, so the statement in the discussion seems internally consistent with the data presented in the paper. Moreover, the authors include pVL and CD4 comparisons in their rebuttal letter, but indicate that these are "for reference during peer review only". I recommend that the authors incorporate these data into their paper.

3. The original reviewer pointed out that the tree presented in S1 is confusing because the oldest sequences are farthest from the root (and vice-versa). The authors reply that, since this is a ML tree with no outgroup included, "the topology does not really reflect the temporal dynamics of the evolution of CRF07_BC". While I appreciate that no temporal information was used during phylogenetic inference, presenting this tree with the root placed in the present position is unnecessarily confusing for the reason the original reviewer identified. If one presents a rooted tree, this inherently suggests that the placement of the root is meaningful, so it's misleading to present a root but then say that it should essentially be disregarded. One possibility is to

present an unrooted tree, but I appreciate that these can be very difficult to interpret. Another possibility is to re-infer a tree using an outgroup, but I appreciate that this would change the topology and therefore the clustering analysis. The authors could therefore consider rooting the existing tree using evolutionary placement algorithms, or even rooting it elsewhere, eg. at the earliest sampled sequence or perhaps even at the midpoint to generate a root that is not visually confusing. Regardless, the authors should state how the tree was rooted.

4. The original reviewer requested that the sentence regarding how the results could potentially be used by public health to target prevention service be clarified. The authors responded by removing the example of "aggressive treatment" as a possible relevant strategy (which is appropriate, since early treatment is standard-of-care regardless of what the infecting strain is). But, removal of this statement leaves no concrete examples of how the study's findings could be used to inform public health decision making, and in fact it is not clear how study findings could inform prevention strategies. The authors may wish to revisit this statement completely or suggest another example (e.g. prioritizing MSM populations for test and treat initiatives?)

Additional comments

1. The manuscript contains confusing and potentially contradictory claims regarding the relationship between disease progression, transmissibility, virulence, replication capacity and disease progression, and the implications of this for HIV spread. The authors cite a study by Cheng et al who showed that CRF07_BC have enhanced transmission capability over other subtypes due to unique genetic features in p6Pol (PTAP insertion and p6d7 deletion). However, the increased transmission capability is claimed to be due to the fact that these genetic features serve to SLOW disease progression, thereby increasing the pool of people who are able to transmit CRF07_BC to others. In the present study however the authors show that the K28E32 subvariant has higher in vitro replication capacity, and data in the rebuttal letter also suggest that it is associated with markers of more SEVERE disease (higher pVL, low CD4), and the authors suggest that these attributes may make this subvariant more transmissible. The authors should address this discrepancy in the discussion.

2. It is unclear how the statement "HIV-1 mutations in RT are often associated with drug resistance" in the importance paragraph is relevant to this study as the identified mutations do not confer resistance. Are the authors suggesting that strains that have higher in vitro replication capacity may have higher potential to develop drug resistance? If so, this is not a good argument, as there is no reason to think that ARVs would not suppress the K28E32 strains as potently as other CRF07 strains. Authors should remove this sentence from the importance statement.

2. line 219 - "indicating" should be changed to "suggesting" when referring to the KE variant's effects on viral replication capacity

3. add "in vitro" before "replication ability", at minimum in abstract, importance and discussion to make it clear that these features were evaluated in vitro.

4. In Figure 2, authors looked for the presence of the 5 RT mutations in *representative* sequences from other subtypes. This is fine for figure presentation, but the authors should also look for these mutations in all available sequences in LANL, not just the small subset of representative sequences. For example ~20% of subtype A2 strains and ~5% subtype D strains carry T338S. This can simply be mentioned in the text, no need to change the figure.

5. Typos in Figure S1 legend: C12 is listed twice, but the second time should be C72. Also, the clusters are called "T" (e.g. T4) in Figure 1 but "C" (e.g. C4) in Figure S1. This is confusing and should be standardized.

Staff Comments:

Preparing Revision Guidelines

For complete guidelines on revision requirements, please see the journal Submission and Review Process requirements at <https://journals.asm.org/journal/Spectrum/submission-review-process>. **Submissions of a paper that does not conform to**

Microbiology Spectrum guidelines will delay acceptance of your manuscript. "

Please return the manuscript within 60 days; if you cannot complete the modification within this time period, please contact me. If you do not wish to modify the manuscript and prefer to submit it to another journal, please notify me of your decision immediately so that the manuscript may be formally withdrawn from consideration by Microbiology Spectrum.

Response to the reviewer's comments

Reviewer comments:

Reviewer #1 (Public repository details (Required)):

It appears that the paper is compliant with this requirement. Most of the data were downloaded from public repositories but the methods describes genotypes from 207 individuals collected with written informed consent, and a range of 207 GenBank accession numbers are provided.

Authors: No need to answer.

Reviewer #1 (Comments for the Author):

Han et al combine molecular epidemiology, phylogenetics and molecular biology approaches to characterize, and investigate potential viral genetic features underpinning the rapid spread of a novel K28E32 variant of CFR007_BC in China. The presented analyses are well undertaken. The authors have also addressed many of the comments provided by a previous reviewer, and these changes have improved the manuscript.

However, some of the original reviewer's comments have not been adequately addressed.

Authors: We thank the reviewer's comments. We further revised manuscript according to these new comments.

Specifically:

1. As the original reviewer correctly pointed out, there are no data in the present paper that show that the K28E32 variant is more transmissible. The authors provide data supporting this variant as having a higher replication capacity and superior RT activity in vitro, but this does not necessarily equate to increased transmissibility between hosts. The original reviewer also correctly emphasized that founder effects, followed by subsequent dissemination of this new variant in MSM as a result of factors that are completely unrelated to viral genetics, cannot be ruled out as the main reason for this variant's spread in this new population. ON the latter point, the original reviewer emphasized that founder virus effects should be emphasized *from the start* (see discussion point #1 in original reviews).

The authors indicates that they have "toned down" their claims, but the manuscript still contains statements claiming that their data support transmissibility, as well as statements that claim that viral genetics are the underlying mechanism for this strain's spread, without adequately acknowledging founder virus effects from the outset. These statements must be revised, in particular the abstract, importance and concluding paragraph of the study. They include:

-> abstract "This variant... more likely formed transmission clusters and drive the rapid expansion of CFR07_BC....". The authors should avoid this causative language, as it is not supported by the data. Instead say, e.g. "the variant.. was overrepresented in transmission clusters, suggesting that it could have driven the rapid expansion of CRF07_BC in MSM, though founder effects cannot be ruled out"

-> importance statement "This study identified five crucial mutations that answers why CRF07_BC had a rapidly increasing prevalence". The authors must remove strong causation claims such as this.

-> lines 178-180: "The generation of the C28E32 variant was mainly involved in the adaption of CRF07_BC to individuals with reported homosexual risk". This strong causative statement is not supported by the data. These types of statements must be toned down and founder effects must be acknowledged as an alternative explanation from the beginning.

-> line 358 "This indicates that co-receptor tropism does not contribute to the rapid spread of K28E32 in MSM". This cannot be concluded from the data. Change to "This suggests"

-> line 370-372 "We identified...a new CRF07_BC variant that.....has a transmission advantage over wild-type". This statement is not supported by the data.

-> lines 372-374 in the final concluding paragraph of the paper state that founder effects could explain their observations, but this statement comes out of the blue with no context, as if it was simply added to address the reviewer concern. As such, it reads like a direct contradiction of the prior sentence (which claims that the virus has a transmission advantage), rather than an explicit acknowledgement that this could represent an alternative mechanism. The final sentence further adds to the confusion as it seems to walk back the transmission statements as being far from conclusive (i.e. "any innate transmission advantage that these viruses have") and in fact the authors concede in their rebuttal letter that future directions include investigating whether this virus is more transmissible.

Authors: We thank the reviewer for further suggestions on this point, and some language editing. We further revised the manuscript to tone down and/or remove the claims of higher transmissibility of the new K28E32 variant, and revised all above-mentioned sentences according to the reviewer's suggestions. In addition, to avoid strong causative statement, we also revised the title as "A new HIV-1 K28E32-RT variant associated with the rapid expansion of CRF07_BC among men who have sex with men" ("causes" to be removed).

We deleted all related descriptions on higher transmissibility of the new K28E32 variant from the revised manuscript. They include but not limited to:

Abstract: "..., more likely formed transmission clusters,..."

Importance: "..., answers why CRF07_BC had a rapidly increasing prevalence and was becoming the predominant strain among MSM."

Introduction: "..., which explain why CRF07_BC had a rapidly increasing prevalence"

Discussion: para. 2: "..., and might be more transmissible at least among MSM." and "...highly transmissible..."

Para.4: "..., more often formed..."

Para.5: "...highly transmissible..."

Last para. "...and has a transmission advantage over..."

With regard to the founder effects, we mentioned and discussed it in para. 2 of Discussion. In addition, a related sentence in the concluding paragraph was also revised: "The emergence and subsequent predominance of the K_{28E32} variant among MSM could be ascribed to its higher in vitro replication ability and/or simply a founder effect of this..."

Even though we toned down and removed the related claims on transmissibility from the paper, we still want to **express our views**:

"the epidemiological data showed an increased prevalence of the K28E32 variant among all high risk groups (MSM, heterosexuals, and even IDUs)(Figure 5), and the Bayesian analysis also showed a rapid expansion of the K28E32 variant accompanied with a decrease of the wild type during the same period (2004-2019) (Fig. 4). With regard to the founder virus effects, however, the wild type was also circulating

among MSM with a very low prevalence, and reversely the K28E32 variant was also circulating among IDUs with a very low prevalence (Fig. 5A). It is interesting that the founder virus effects did not cause a rapidly increased prevalence of the K28E32 variant among IDUs as observed among MSM, and also not cause a rapidly increased prevalence of the wild type in MSM as observed among IDUs (maybe the founder virus effects can be well used for explaining a result, but be difficult to predict an expansion of a new variant)."

2. The original reviewer pointed out that evidence linking the K38E32 variant to correlates of HIV disease progression would strengthen the study. The revised manuscript (lines 360-369) indicates that this was not possible because "sequences were downloaded from HIV database and related clinical information were unavailable". However the methods section describes the recruitment of 207 participants with HIV with written informed consent who underwent HIV genotyping, for whom sociodemographic (and presumably clinical) data were also collected, so the statement in the discussion seems internally consistent with the data presented in the paper. Moreover, the authors include pVL and CD4 comparisons in their rebuttal letter, but indicate that these are "for reference during peer review only". I recommend that the authors incorporate these data into their paper.

Authors: First, in this study, most sequences were downloaded from HIV database, and 207 sequences were obtained from HIV-1 infected individuals. The related clinical information of downloaded were unavailable. For the 207 HIV-1 infected participants who were recruited in a cross-sectional investigation, their sociodemographic data did not include clinical data (CD4 count and VL). The clinical data will be available only when they visit special hospital for receiving ART.

Second, we are sorry for some ambiguous description on the result in Fig. R2 in our previous rebuttal. The data presented in Fig. R2. were obtained from a previously published paper (*Retrovirology* 2021, 18:22) by my former collaborator (Dr. Jin Zhao at Shenzhen CDC)(Please see Fig. R1 below). The data was not same to the new data (sequences obtained from 207 individuals without clinical information of VL and CD4 count) in this study. In Dr. Zhao's paper, they compared the CD4 count and HIV-1 VL among MSM infected with three different HIV-1 CRFs (CRF01_AE, CRF07_BC, and CRF55_01B). To answer the question by the previous reviewer, we asked Dr. Zhao for the original data of CRF07_BC (including sequences, CD4 count and VL), to re-analyze and compare the CD4 count and VL between the wild type and the new K28E32 variant of CRF07_BC. Dr. Zhao provided the data but did not agree us to independently publish the data. In addition, as we mentioned previously, the result was very preliminary and not solid since the sample size was small especially for the wild-type strains (9 wild-type vs. 421 K28E32 variants in the VL analysis, and 22 wild-type vs. 791 K28E32 variants in the CD4 count analysis). We will collaborate with Dr. Zhao and other teams to investigate this issue by collecting more sequence and clinical data. It is the main reason why we said that "*Our preliminary unpublished analysis is just for reference during this peer review*".

Fig R1. The data presented in the *Retrovirology* paper by Dr. Jin Zhao.

3. The original reviewer pointed out that the tree presented in S1 is confusing because the oldest sequences are farthest from the root (and vice-versa). The authors reply that, since this is a ML tree with no outgroup included, "the topology does not really reflect the temporal dynamics of the evolution of CRF07_BC". While I appreciate that no temporal information was used during phylogenetic inference, presenting this tree with the root placed in the present position is unnecessarily confusing for the reason the original reviewer identified. If one presents a rooted tree, this inherently suggests that the placement of the root is meaningful, so it's misleading to present a root but then say that it should essentially be disregarded. One possibility is to present an unrooted tree, but I appreciate that these can be very difficult to interpret. Another possibility is to re-infer a tree using an outgroup, but I appreciate that this would change the topology and therefore the clustering analysis. The authors could therefore consider rooting the existing tree using evolutionary placement algorithms, or even rooting it elsewhere, eg. at the earliest sampled sequence or perhaps even at the midpoint to generate a root that is not visually confusing. Regardless, the authors should state how the tree was rooted.

Authors: We agree with the reviewer. As suggested, we rooted the ML tree using midpoint rooting method. The new tree (new supplementary Fig. S1) showed similar topology with the ML tree presented in Fig. 3 of the paper.

4. The original reviewer requested that the sentence regarding how the results could potentially be used by public health to target prevention service be clarified. The authors responded by removing the example of "aggressive treatment" as a possible relevant strategy (which is appropriate, since early treatment is standard-of-care regardless of what the infecting strain is). But, removal of this statement leaves no concrete examples of how the study's findings could be used to inform public health decision making, and in fact it is not clear how study findings could inform prevention strategies. The authors may wish to revisit this statement completely or suggest another example (e.g. prioritizing MSM populations for test and treat initiatives?)

Authors: We thank the reviewer for the suggestion. We revised this sentence as "The results could allow public health officials to use this marker (especially E28K and K32E mutations in RT coding region) to target prevention measures prioritizing MSM population for test and treat initiatives".

Additional comments

1. The manuscript contains confusing and potentially contradictory claims regarding the relationship between disease progression, transmissibility, virulence, replication capacity and disease progression, and the implications of this for HIV spread. The authors cite a study by Cheng et al who showed that CRF07_BC have enhanced transmission capability over other subtypes due to unique genetic features in p6Pol (PTAP insertion and p6d7 deletion). However, the increased transmission capability is claimed to be due to the fact that these genetic features serve to SLOW disease progression, thereby increasing the pool of people who are able to transmit CRF07_BC to others. In the present study however the authors show that the K28E32 subvariant has higher in vitro replication capacity, and data in the rebuttal letter also suggest that it is associated with markers of more SEVERE disease (higher pVL, low CD4), and the authors suggest that these attributes may make this subvariant more transmissible. The authors should address this discrepancy in the discussion.

Authors: First, as mentioned by the reviewer, the paper by Cheng et al suggested CRF07_BC have enhanced transmission capability over other subtypes due to the genetic features in p6Gag (PTAP insertion and p6d7 deletion). They concluded an association between CRF07_BC infection and quick growth of HIV-1 transmission clusters (in other word, higher transmission growth index than other subtypes/CRFs) based on transmission cluster analyses of *pol* sequences (**rather than p6-covering gag region: p6Gag**). In other words, they were unable to analyze and determine the proportion of the variants with p6Gag mutation (insertion and/or deletion) in large CRF07_BC transmission clusters. Of importance is that there also lacked epidemiological and Bayesian population expansion dynamics data of the variants with p6Gag mutation to show the origin, evolution and expansion of the p6Gag variants. Therefore, the association between higher transmissibility of CRF07_BC and the genetic features in p6Gag was rather weak. On the other hand, Cheng et al said in the results "To strengthen the robustness of the analysis, sensitivity analysis restricted to HIV infected MSM also confirmed the association between CRF07 BC infection and quick growth of HIV-1 transmission clusters (P, 0.001; Table S1 in the supplemental material)". Similarly, they were also unable to determine whether the vast majority of CRF07_BC strains circulating among MSM carry p6Gag mutations. According to our analysis, however, **the vast majority of CRF07_BC strains appearing in large CRF07_BC transmission clusters and/or circulating among MSM belonged to the K28E32 variants**. Therefore, we said "However, the p6Δ7 variant did not explain the rapidly growing prevalence of CRF07 BC among MSM" in Introduction of our paper.

Second, indeed, the association of CRF07_BC infection with relatively lower VL but slightly higher CD4 count (compared to infections with other HIV-1 subtypes/CRFs) was observed in some studies (e.g. Cheng's paper and Fig. 2 in the *Retrovirology* paper I mentioned above). Cheng et al gave a possible explanation that SLOW disease progression increases the pool of people who are able to transmit CRF07_BC to others (maybe it is just an explanation); however, a large number of studies (evidences) supported that higher VLs lead to higher HIV transmission rates, and VL is the chief predictor of the risk of HIV-1 transmission (representative publications: N Engl J Med. 2000 Mar 30;342(13):921-9; AIDS 1999 Jul 30;13(11):1377-85; AIDS. 2014 Apr 24; 28(7): 1021–1029). It is also why ART is so much important to prevent HIV-1 transmission.

In addition, in our study, we focused on comparing the new K28E32 variant with the wild-type (E28K32) of CRF07_BC strains, which is different from Cheng's and other papers that compared CRF07_BC as a whole with other HIV-1 subtypes/CRFs. We did not deny that SLOW disease progression (slightly lower VL and higher CD4 count) might be associated with a more rapidly increasing prevalence of CRF07_BC than other HIV-1 CRFs (e.g. CRF01_AE) (as seen in MSM)(although the mechanism still needs to be investigated, some previous studies suggested it might be involved in the R5 tropism of almost all CRF07_BC strains since R5 strains appeared to be advantage over X4 strains for HIV-1 sexual transmission). However, given the facts that the vast majority of CRF07_BC circulating in MSM in Shenzhen belonged to the K28E32 variants (in this study, please also see the answer to the question 2 in the major comments) and the prevalence of CRF07_BC was firstly observed to exceed CRF01_AE among MSM in Shenzhen (our previous paper in Sci Report. 2016, 6:28703.), we suggest that higher *in vitro* replication capacity of the new K28E32 variant was associated with (or contributed to) the rapid expansion of CRF07_BC among MSM (the main conclusion of this study).

2. It is unclear how the statement "HIV-1 mutations in RT are often associated with drug resistance" in the importance paragraph is relevant to this study as the identified mutations do not confer resistance. Are the authors suggesting that strains that have higher *in vitro* replication capacity may have higher potential to develop drug resistance? If so, this is not a good argument, as there is no reason to think that ARVs would not suppress the K28E32 strains as potently as other CRF07 strains. Authors should remove this sentence from the importance statement.

Authors: As suggested, we removed the sentence from the revised manuscript.

2. line 219 - "indicating" should be changed to "suggesting" when referring to the KE variant's effects on viral replication capacity

3. add "in vitro" before "replication ability", at minimum in abstract, importance and discussion to make it clear that these features were evaluated *in vitro*.

Authors (questions 2-3): We thank the reviewer for these important suggestions. We updated them in the revised manuscript.

4. In Figure 2, authors looked for the presence of the 5 RT mutations in *representative* sequences from other subtypes. This is fine for figure presentation, but the authors should also look for these mutations in all available sequences in LANL, not just the small subset of representative sequences. For example ~20% of subtype A2 strains and ~5% subtype D strains carry T338S. This can simply be mentioned in the text, no need to change the figure.

Authors: As suggested, we further analyzed subtype A (including sub-subtypes A1, A2, A3, A4, A6, A7, and A8), and subtype D, and added the new results in the supplementary Table S?, and mentioned in the Result section.

5. Typos in Figure S1 legend: C12 is listed twice, but the second time should be C72. Also, the clusters are called "T" (e.g. T4) in Figure 1 but "C" (e.g. C4) in Figure S1. This is confusing and should be standardized.

Authors: Thank the reviewer for pointing out these typos. We corrected them in the revised manuscript.

September 15, 2022

Prof. Chiyu Zhang
Shanghai Public Health Clinical Center, Fudan University
2901 Cao Lang Road, Jinshan District, Shanghai, China
Shanghai
China

Re: Spectrum02545-22R1 (A new HIV-1 K28E32-RT variant associated with the rapid expansion of CRF07_BC among men who have sex with men)

Dear Prof. Chiyu Zhang:

Please address the issue of Ref #32 pointed out by the reviewer below.

Your manuscript has been accepted, and I am forwarding it to the ASM Journals Department for publication. You will be notified when your proofs are ready to be viewed.

Sincerely,

Takamasa Ueno
Editor, Microbiology Spectrum
